# Antimicrobial use in laboratory rodent facilities in Australia and New Zealand- a cross-sectional survey of veterinarians and facility managers

**Rebbecca S. Wilcox** ⬦ * ◐ , **Marc S. Marenda**‡, **Joanne M. Devlin**‡, **Colin R. Wilks**◐

Asia Pacific Centre for Animal Health, Melbourne Veterinary School, Faculty of Science, The University of Melbourne, Parkville, Victoria, Australia

◐ These authors contributed equally to this work.
‡ MSM and JMD also contributed equally to this work.
* rebbecca.wilcox@rmit.edu.au

**Data Availability Statement:** All relevant data are within the paper and its Supporting Information files.

**Funding:** The author(s) received no specific funding for this work.

## Abstract

This cross-sectional study surveyed veterinarians and facility managers to characterise the use of antimicrobials in laboratory rodent facilities within Australia and New Zealand. Most facilities (71%) reported routine administration of antimicrobials. The indications for antibiotic use reflected those described in publications and differed significantly to reasons for use in non-laboratory animals. Antimicrobials used include those of critical importance to human health, and access to these drugs is unregulated, as prescription-only classes are ordered through research catalogues, without human or veterinary physician prescriptions. The ways in which antimicrobials are used in Australian and New Zealand rodent facilities are likely contributing to antimicrobial resistance within rodent populations, particularly as they are largely administered in drinking water, risking subtherapeutic dosing. Much antimicrobial use reported is unnecessary and could be replaced with changes to husbandry and handling. The generation of resistance in both pathogenic and commensal microbes may also represent a work health and safety issue for humans working with these animals. Reported disposal of antimicrobials included discharge into wastewater, without inactivation, and some respondents reported disposal of substrate, or soiled bedding, nesting material, and disposable enrichment items, from treated animals and medicated feed into landfill, without prior inactivation. Environmental contamination with resistant microbes and antimicrobials is a significant driver of antimicrobial resistance. As such, significant opportunities exist to implement judicious and responsible use of antimicrobials within research rodent facilities in Australia and New Zealand, with a particular focus on instituting aseptic surgery, optimising dosing regimens, and inactivation of medicated water and substrate before disposal.

**Competing interests:** The authors have declared that no competing interests exist.

# Introduction

Antimicrobial drug resistance (AMR) is a public health emergency and existential threat to the advances in modern medicine of the last 80 years. AMR presently accounts for over seven million deaths annually and is projected to cause ten million deaths by the year 2050 [1–3].

Bacterial resistance to antibiotics is inevitablehttps://www.ncbi.nlm.nih.gov/pmc/articles/PMC5250489/ - bib17, both spontaneously in nature, and in response to exposure to antimicrobials. Resistance develops when microbes undergo mutation or exchange resistance genes and is accelerated in the presence of antimicrobials [4]. To slow AMR, selective pressures must be minimised through judicious use of antimicrobials [5].

Importantly, the pipeline for new antimicrobials is slow and has been unfruitful, due to stringent regulatory requirements and low investment returns for pharmaceutical companies. Effective new antimicrobials are used for short durations when compared with drugs for other conditions, and the restricted use of newly approved antimicrobials further reduces the market share for investing companies, along with the inevitable decline in their efficacy and value when AMR develops [6].

A comprehensive 'One Health' approach is essential for successful prevention and management of numerous infectious diseases, including those caused by emerging pathogens and those due to AMR [7, 8]. AMR has been described as the 'quintessential One Health issue' [9].

A One Health approach is essential to addressing AMR, given many human antimicrobials are also used in veterinary medicine. There is evidence that some clinically relevant resistant bacteria and/or their antimicrobial resistance genes (ARGs) readily transfer from bacteria present in animals to those in humans [10, 11]. Recent studies have shown identical AMR signatures in pathogenic *Escherichia coli* isolated from both humans and their companion animals, humans, and other species, and between animals and the environment [12, 13].

Decades of injudicious antimicrobial use, in human and veterinary medicine, and in agriculture have resulted in the development of extensive AMR in pathogenic and opportunistic microbes, resulting in widespread morbidity and mortality. AMR bacteria are not necessarily more virulent, however, delays in infection control due to ineffective initial treatment and testing required to determine appropriate therapy, dramatically increases the cost of medical care, and impacts human and animal patients' health [14].

Whilst the AMR implications of antimicrobial use in agriculture, aquaculture, companion animals and equids are established and the subject of considerable quantitative and qualitative research, there is a knowledge gap with respect to antimicrobial use in laboratory animals, particularly research rodents [15–17]. This is important to understand, given the considerable number of research rodents used worldwide, estimated to exceed 120 million, and their utility in biomedical research, meaning use has only grown with time [18, 19].

Resistant microbes pose significant zoonotic disease risks [20, 21]. Veterinarians must consider the impacts of antimicrobial use, administering them sparingly and appropriately, to safeguard the health benefits to animals, preserve efficacy and minimise promotion of AMR [22]. This also applies to use in research animals, and likewise, must be considered by animal researchers and laboratory animal veterinarians [1].

This study aimed to investigate the use of antimicrobials in research rodent facilities in Australia and New Zealand. Based on anecdotal observations and published research protocols, the authors hypothesised that most research rodent facilities would report current antimicrobial use, with a high prevalence of in-water administration and use of fluoroquinolone, tetracycline, and sulphonamide classes. This cross-sectional study aimed to question these assumptions and characterise the means in which antimicrobial containing substrates were disposed of, including wastewater, food, and bedding. The study aimed to address the

significant knowledge gap around antimicrobial use in laboratory rodents, contributing to a missing component of the One Health paradigm in addressing AMR.

Prior to this survey there were no published data on the extent of antimicrobial use in rodent laboratory facilities, in Australia and New Zealand, or globally.

## Materials and methods

The source population comprised the 158 eligible research rodent facilities across each of the states, territories and islands of Australia and New Zealand, respectively. The total number of eligible facilities was determined using a list provided by the Australia and New Zealand Laboratory Animal Association (ANZLAA) Secretariat. The eligible respondent population comprised veterinarians and animal facility managers, working in these facilities at the time of survey publication.

The study population was comprised of eligible respondents, answering the survey between February 2nd, 2020, to April 2nd, 2020. Respondents self-selected and the survey was distributed online using the Qualtrics™ survey platform, through the ANZLAA members' email list.

The survey questions covered demographic aspects of respondents and the prevalence of antimicrobial use, according to class, route and duration of administration, reasons for use and how antimicrobials are sourced and disposed of. To be confident that the proportion of respondents giving a particular answer was within a constant margin of error of 6.5% of a true prevalence of 50%, a total of ninety-four (94) completed surveys were required. Sample size calculations were conducted with finite population correction, and an assumption of 50% prevalence.

Data were analysed using a live excel calculator, from The Statistical Consulting Centre, University of Melbourne, using a 95% confidence interval for a single population proportion, θ, with a finite population adjustment. The number of required responses was calculated using the Australian Bureau of Statistics Sample Size Calculator.

Data were reported in proportions: the number of respondents selecting an answer or option were divided by the total number of respondents attempting the question. Responses of 'do not know' and unanswered questions were interpreted as missing values and were excluded from analysis. To improve interpretation, or where there was redundancy, some response categories were merged. For some complex questions, respondents provided replies in free text format.

The study was conducted according to the guidelines of the Declaration of Helsinki. The anonymous online survey, recruitment text and accompanying Plain Language Statement were approved by the Human Research Ethics Committee of the Faculty of Veterinary and Agricultural Science, The University of Melbourne, Ethics ID Number 1955621.1. The Ethics Committee approved 'Consent being implied', given, that to complete the survey, participants first read an Ethics approved Plain Language Statement, that explained that all participant responses were anonymous, and could not be re-identified, and that participation was voluntary.

### Data availability

The survey questionnaire and responses are included in the S1 Dataset.

### Inclusivity in global research

Additional information regarding the ethical, cultural, and scientific considerations specific to inclusivity in global research is included in the S1 Text.

## Results

### Survey response rate and respondent occupation and location

Of the source population, comprising 158 eligible rodent facilities, there was a response rate of 60% (95 respondents). Respondents were composed of 46% non-veterinarians with facility managerial roles and 51% veterinarians in non-managerial roles. Three respondents (3%) did not report their roles, though completed the survey, with other answers meeting eligibility criteria.

Eighty-two percent (82%) of respondents were from Australia and 18% from New Zealand (S1 Dataset).

### Facility type and housing

Eighteen percent (18%) of respondents described facilities that met multiple criteria for facility type and affiliation (Fig 1). Most respondents were from universities followed by private institutes. Approximately half were associated with a human hospital, the majority, 60%, reporting facility co-location within a human hospital. A small percentage were government facilities, or animal production facilities or vendors.

Cage systems varied, with 24% using open-top conventional caging, 31% Physical Containment Level (PC) 1, 86% PC2, and 15% PC3, and one third reported multiple caging systems, usually both conventional open-top cages, and PC1 +/- PC2.

### Research disciplines

Most facilities (78%) reported research across multiple disciplines, including studies in: metabolic disease (67%), neuroscience (55%), oncology (53%), cardiovascular disease (40%), the microbiome (38%), infectious disease (37%), biologics and vaccine production (27%), and breeding of rodents for commercial supply (12%). A third reported use for anatomy and physiology teaching purposes.

### Prevalence of antimicrobial use

Seventy-one percent (71%) of rodent facilities reported routinely using antimicrobials, with an identical prevalence across Australian and New Zealand respondents. Facilities associated with human hospitals reported the greatest use of antimicrobials (79%), followed by 76% of university respondents. Approximately two thirds (64%) of private institutes, and those without hospital or university affiliation reported routine use. A third of the commercial animal breeding and supply facilities reported routine use.

### Reasons for use

Most facilities reported multiple reasons for using antimicrobials, listed in Table 1. When used to treat infections, one third of respondents (29%) reported routine use of microbial culture and antimicrobial sensitivity testing (AST) to guide therapeutic decisions.

### Routes of administration

Most respondents reported multiple routes of antimicrobial administration, most commonly in drinking water (70%), commercially formulated chow (37%) or via injection (66%). Administration via commercially formulated chow was common. The least prevalent routes were in-facility bespoke addition of antimicrobials to powdered food, and subcutaneous insertion of antimicrobial releasing devices or minipumps. See Fig 2.

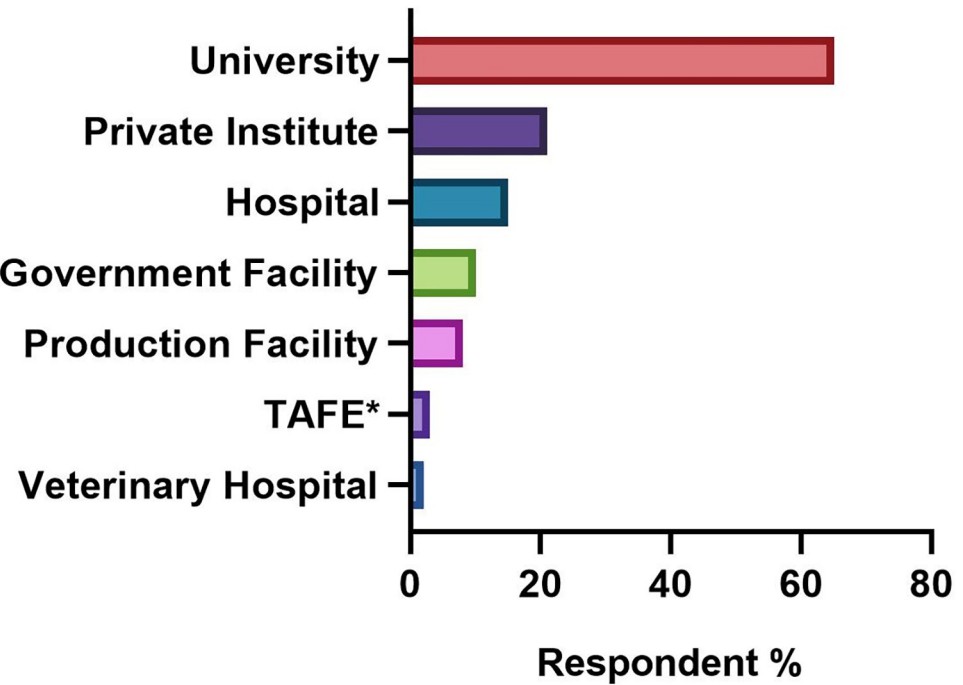

**Fig 1. Types of facilities identified by respondents (n = 95) to a survey exploring use of antimicrobials in research rodents in Australia and New Zealand.** *(TAFE—Technical and Further Education Institution).

## Prevalence of usage according to antimicrobial class

The fluoroquinolone class of antimicrobial was the most administered antimicrobial, with enrofloxacin the most prevalent specific antimicrobial used. Of the 67% of respondents administering enrofloxacin, which is light sensitive, in-water, only 48% reported covering medicated water bottles to prevent light exposure. Prevalence of antimicrobial use by class, route and duration of administration, and treatment of medicated water prior to disposal is listed in **Table 2**.

**Table 1. Reasons for use of antimicrobials identified by respondents (n = 95) to a survey exploring use of antimicrobials in research rodents in Australia and New Zealand.**

| Reason for Use | Percentage of Respondents |
|---|---|
| Treating infections in individual animals | 74% |
| Infection prevention in genetically immune-deficient rodents | 58% |
| Prevention of infection at surgery | 54% |
| Induction of genes (tetracycline promoters) | 51% |
| Infection prevention post irradiation or chemotherapy | 45% |
| Treatment of infections in rodent colonies | 31% |
| Alteration of the microbiome | 25% |
| Testing of antimicrobials | 8% |

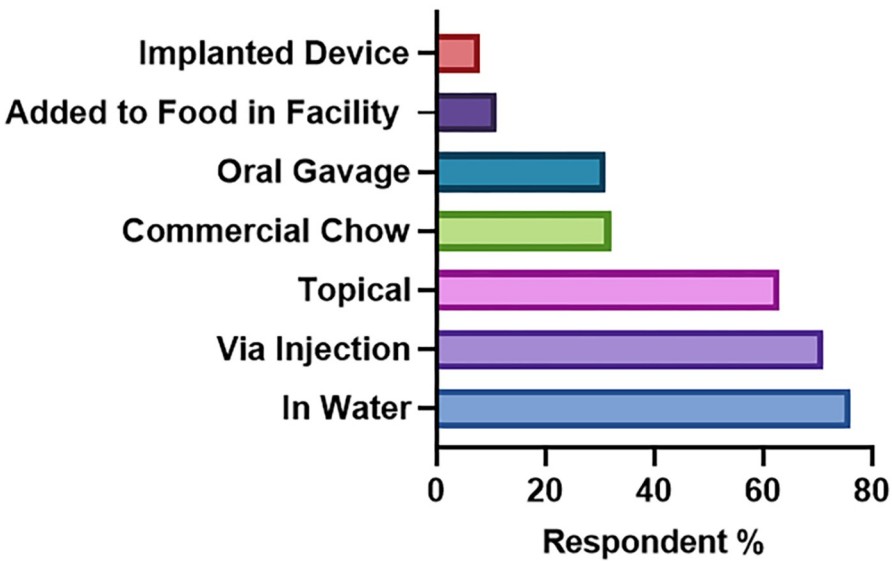

**Fig 2. Route of administration of antimicrobials identified by respondents (n = 95) to a survey exploring use of antimicrobials in research rodents in Australia and New Zealand.**

## Co-administration of antimicrobials

Twenty-nine percent (29%) of respondents reported routinely co-administering multiple antimicrobials. Given the multitude of combinations, free-text responses were requested, detailing specific antimicrobial combinations. Examples of the free text responses are shown in **Fig 3**.

## Disposal of antimicrobials

Most facilities (81%) reported disposing medicated water, untreated, down the drain or sink. Those inactivating the antimicrobials within the water reported using chemical or heat inactivation. A minority engaged specialist contractors to dispose of water containing antimicrobials, see **Fig 4**.

For antimicrobial treated rodent carcasses and their substrate, multiple means of disposal were reported by individual facilities. Differing physical containment designations, corresponding to degrees of biosecurity, of which most respondents had two or more designations, have specific legal obligations regarding disposal, accounting for a range of disposal methods.

Respondents reported disposal of treated animal carcasses as follows: autoclaving or not autoclaving carcasses followed by disposal via a medical waste contractor, 35% and 80% respectively; 4% via incineration, 2% via landfill without prior autoclaving and 2% of respondents chose not to answer. Bedding and substrate from treated animals were disposed of via a medical waste contractor (76%), sent to land fill with no autoclaving (31%), or incinerated (2%), whilst 2% preferred not to answer.

Approximately one third of respondents (27%) reported disposing of medicated chow to land fill without treatment. Most respondents used multiple disposal methods including specialised contractors with or without prior autoclaving (87%) and incineration (3%) as per **Fig 5**.

**Table 2. Prevalence of antimicrobial use reported by respondents (n = 95) from rodent research facilities in Australia and New Zealand, according to class, route, duration of treatment, and inactivation treatment of medicated water pre-disposal.**

| Antimicrobial Class | Proportion of Facilities | Route of Administration | Duration < 7 days | Duration > 7 days | Inactivation of Water Before Disposal |
|---|---|---|---|---|---|
| Fluoroquinolones e.g., enrofloxacin | 79% | Injection– 74% | 63% | 49% | 19% |
|  |  | In water– 67% |  |  |  |
|  |  | Gavage/chow- 24% |  |  |  |
| Tetracyclines e.g., doxycycline | 61% | Chow* – 52% | 21% | 79% | 18% |
|  |  | In water– 39% |  |  |  |
|  |  | Injection– 18% |  |  |  |
| Neomycin | 56% | Topical– 61% | 52% | 48% | 10% |
|  |  | In water– 39% |  |  |  |
|  |  | Oral *– 10% |  |  |  |
| Trimethoprim-sulphonamides | 52% | In water– 52% | 67% | 50% | 0 |
|  |  | Oral– 30% |  |  |  |
|  |  | Injection– 44% |  |  |  |
| Amphenicols e.g., chloramphenicol | 39% | Topical– 100% | 95% | 10% | N/A |
| Penicillins, e.g., amoxicillin, benzyl penicillin | 36% | In water (amoxicillin)– 45% | 53% | 58% | 0 |
|  |  | Injection (benzyl penicillin)– 75% |  |  |  |
| Polymyxins | 35% | Topical -83% | 88% | 35% | 0 |
|  |  | In water– 28% |  |  |  |
|  |  | Injection– 11% |  |  |  |
| Glycopeptides e.g., vancomycin | 13% | In water– 57% | 33% | 67% | 33% |
|  |  | Gavage– 43% |  |  |  |
|  |  | Injection– 29% |  |  |  |
| Metronidazole | 19% | In water– 20% | 50% | 50% |  |
|  |  | Oral– 60% |  |  |  |
|  |  | Injection– 30% |  |  |  |
| Potentiated penicillins, e.g., amoxicillin-clavulanate | 19% | Injection– 80% | 78% | 30% | 0 |
|  |  | In water– 30% |  |  |  |
|  |  | Oral– 20% |  |  |  |
| Cephalosporins, e.g., cefovecin | 18% | Injection (3rd generation)– 80% | 89% | 11% | N/A |
|  |  | Oral (1st generation)– 30% |  |  |  |
| Aminoglycosides | 11% | In water– 50% | 60% | 40% | 0 |
|  |  | Injection– 50% |  |  |  |
|  |  | Oral– 17% |  |  |  |
| Antifungal azoles, e.g., itraconazole | 11% | In water– 17% | 60% | 40% | 0 |
|  |  | Oral—17% |  |  |  |
|  |  | Topical -67% |  |  |  |
| Macrolides, e.g., erythromycin | 4% | In water– 50% | 0 | 100% | 0 |
|  |  | Oral– 50% |  |  |  |
| Lincosamides, e.g., clindamycin | 2% | Injection– 50% | ** | ** | N/A |
|  |  | Oral gavage– 50% |  |  |  |
| Amphotericin B | 2% | In water– 100% | ** | ** | 0 |

*Oral refers to oral gavage, except for tetracyclines and trimethoprim sulphonamides, where it is mostly administered in commercially compounded rodent chow

** not answered

*"Polymyxin B sulphate + neomycin sulphate or enrofloxacin oral solution and amoxicillin in drinking water, post-irradiation."*

*"Metronidazole + amoxicillin in chow...amoxicillin + clarithromycin + metronidazole + omeprazole combination in feed for Helicobacter treatment in immunodeficient mice."*

*"...ampicillin + vancomycin + imipenem + metronidazole + ciprofloxacin in water (to reduce gastrointestinal biome)."*

*"...vancomycin, ampicillin, neomycin in oral gavage."*

*"Vancomycin, metronidazole, gentamicin, kanamycin, colistin, cefaclor, erythromycin in drinking water."*

*"Trimethoprim sulphonamide + oxytetracycline in food."*

*"...cephalexin and enrofloxacin."*

**Fig 3. Free text responses describing co-administered antimicrobials reported by respondents (n = 95) from rodent research facilities in Australia and New Zealand.**

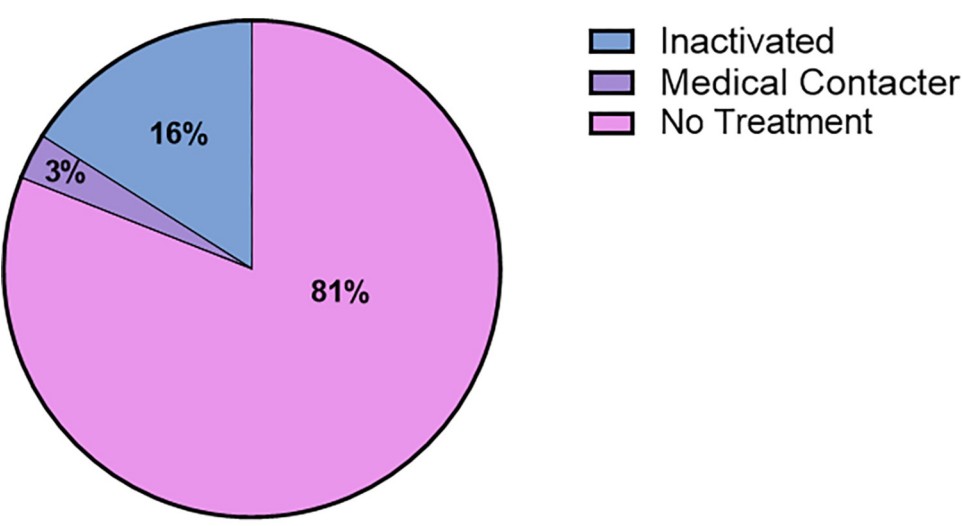

**Fig 4. Management of water containing antimicrobials before disposal, reported by respondents (n = 95) to a survey exploring use of antimicrobials in research rodents in Australia and New Zealand.**

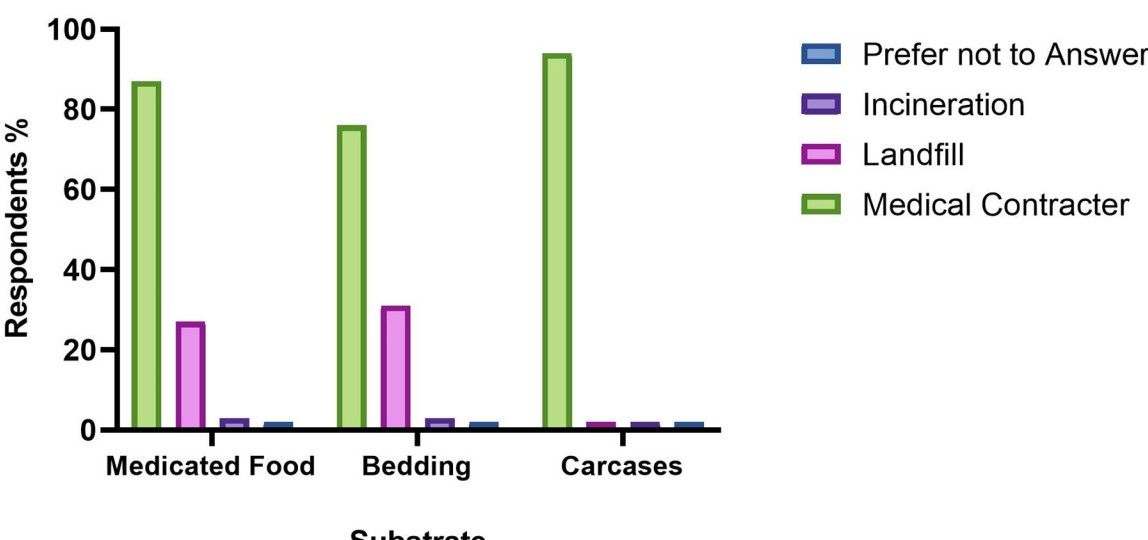

**Fig 5. Methods of disposal of antimicrobial containing substrate: Medicated animal carcasses, their bedding and medicated food according to respondents (n = 95) to a survey exploring use of antimicrobials in research rodents in Australia and New Zealand.**

### Sourcing of antimicrobials

Most facilities reported sourcing antimicrobials through multiple channels, including veterinary wholesalers (93%), research chemical and reagent suppliers (41%), with a smaller number sourcing from livestock food stores and online animal supply stores (4%), private veterinarians (4%), compounding chemists (2%) and via food compounders with a veterinary script (2%), see **Fig 6**.

In most cases multiple persons within facilities administered antimicrobials, including animal technicians in 86% of facilities, researchers in 84%, and veterinarians in 75%. The use of prescription-only antimicrobials was authorised by veterinarians in 83% of facilities, followed by authorisation either or also via the Institutional Drugs and Poisons License Holder, (34%). In 17% of facilities, prescription antimicrobials were used without veterinary prescription or direction, whilst 6% of respondents reported that the Animal Ethics Committee (AEC) authorised their use.

### Standard Operating Procedures (SOPs) for use and disposal of antimicrobials

Over half of facilities (60%) had standardised protocols or SOPs, for the use of antimicrobials. The remaining facilities did not (34%) or chose not to answer (6%). Twenty-four percent (24%) of facilities had SOPs for disposal of antimicrobials, and 75% did not. One percent (1%) of respondents chose not to answer.

### Discussion

This survey shows that a wide range of antimicrobials are used in most rodent facilities in Australia and New Zealand. They are used across a range of research types, and include several

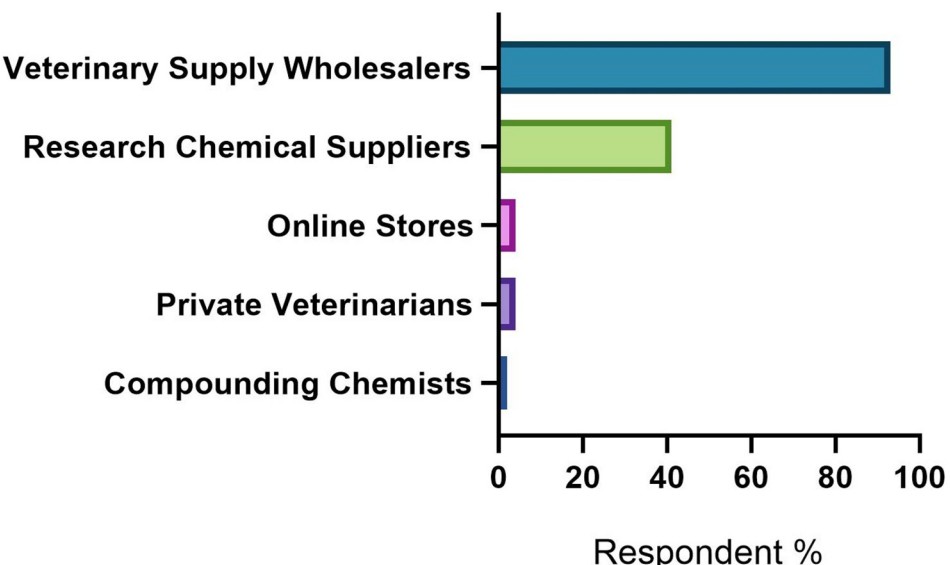

**Fig 6. Sources of antimicrobials identified by respondents (n = 95) to a survey exploring use of antimicrobials in research rodents in Australia and New Zealand.**

antimicrobial classes, some of which are of critical importance to human health. The survey also highlights that the common routes of administration and dosing may be inappropriate, and in general, inadequate measures are taken to ensure the antimicrobials are not released in discarded solids and wastewater.

## Most laboratory rodent facilities in Australia and New Zealand use antimicrobials

The survey revealed a remarkable consistency of antimicrobial use in rodent facilities, across both Australian and New Zealand, with an identical prevalence of 71%. This geographic homogeneity was reflected across types of research reported, and classes of antimicrobials used. This data cannot be compared with that from other regions, as there are no similar surveys or published data, to the authors' knowledge. However, anecdotally, and as detailed in literature on laboratory animal medicine and that describing contemporary research protocols, there is significant antimicrobial use, at unquantified levels [23–27].

## Antimicrobials used include those of critical importance to human health and access is unregulated

The survey documented administration of an extensive range of antimicrobial classes which differs from that used by veterinarians in treating companion or agricultural animal species. The Australian Strategic and Technical Advisory Group on Antimicrobial Resistance (ASTAG) has rated antimicrobials used in humans, companion, and food production animals, and those used in both, according to their importance in treating infections, and the severity of consequences should resistance emerge or be amplified [28]. Antimicrobial use in the laboratory animal sector is not included within ASTAG reviews [28].

There are no regulations for use in laboratory rodents and no antimicrobials registered for rodents. All use is off label.

Several antimicrobials not used routinely in veterinary medicine, including those requiring governmental authorisation in local human settings, such as vancomycin, colistin and ciprofloxacin, (enrofloxacin being the veterinary drug that is metabolised to ciprofloxacin) are used commonly in laboratory rodents, in Australia and New Zealand. Vancomycin and colistin are accessed through research supply catalogues without prescription. In Australia and New Zealand, a veterinarian must prescribe Schedule 4 (prescription animal remedy) antimicrobials for companion and food animals, whereas this is not required for use in laboratory rodents [29].

## Antimicrobial use concurs with that described in published literature

The reasons for antimicrobial use reported in the survey concur with published research protocols and laboratory animal literature.

## Infection prophylaxis in immune deficient genetically engineered mouse models

The survey results mirror numerous publications detailing husbandry of immune deficient colonies, with in-water administration of prophylactic antimicrobials, primarily enrofloxacin, sulphonamides, and amoxycillin-clavulanate [1, 30, 31]. In Australia and New Zealand, enrofloxacin was the most common choice, followed by trimethoprim-sulphonamide.

## Prophylaxis around time of surgery

Prophylactic use around surgery was reported commonly in the survey. Experimental surgery on rodents is seldom performed by veterinarians or human surgeons. Surgeries are mostly undertaken by researchers with little to no formal training in aseptic technique [32]. Lack of training may unnecessarily inflate rates of post-operative infections and training in aseptic technique could obviate such routine use [33, 34].

## Antibiotic induction or silencing of gene expression

Tet-On/Tet-Off mice are a commonly used mouse model in which a gene expression system has been inserted in the mouse genome. Gene expression can be turned 'on' or 'off' with tetracyclines in a mouse's food or water. A PubMed search with the term "Tet-on and Tet-off mice," returns over sixty-four million results, the model being used abundantly in biomedical research since its introduction in 1992 [35]. Extensive use of this model in Australia and New Zealand, as reported, coincides with the global literature.

## Prophylaxis in whole body radiation and bone marrow transplantation

Bone marrow transplantation is commonly performed in mice, wherein they are subjected to whole body ionising irradiation, allowing subsequent engraftment of donor bone marrow. Post-transplantation, mice are immune compromised. Radiation damages the intestinal epithelium, permitting intestinal bacteria to translocate into the bloodstream causing high morbidity and mortality. Likewise, environmental microbes and pathogens cause opportunistic infections. Oral antimicrobials prior to irradiation are used prophylactically in humans, as standard of care [36, 37]. Recent studies have demonstrated in mice that similar reductions in infection are achieved with strict asepsis and use of acidified water, rather than antimicrobials [38].

Global publications cite in-water or gavaged administration of metronidazole, neomycin, ciprofloxacin and tetracyclines as prophylaxis [38]. Whilst the survey found high prevalence of use in irradiated mice, enrofloxacin (precursor of ciprofloxacin) was used most frequently and usually administered in water or by injection, not gavage.

## Induction of microbiome dysbiosis

Microbiome research, the study of the complex consortia of microorganisms, their genes, and interactions with the host is dramatically changing our understandings of medicine [39]. There is growing interest in manipulation of the microbiome in mouse models to study its effects on health and disease. This is mostly achieved with antimicrobials [40, 41]. A PubMed search on "microbiome models in mice, using antimicrobials" returns over five million results.

Most published antibiotic-induced gut microbial imbalance or dysbiosis studies employ so-called "cocktails" comprising combinations of vancomycin, neomycin, ampicillin, metronidazole, polymyxins such as colistin sulphate, carbapenems, third generation cephalosporins and other drugs [40]. One representative example prescribes: 6–8 weeks of ampicillin plus sulbactam, vancomycin, ciprofloxacin, imipenem, and metronidazole administered *ad libitum* in the drinking water to mice [42]. Most studies describe in-water administration [40, 43].

The data from this survey mirror such protocols, with respondents reporting administration of at least three antimicrobials concurrently and in-water administration being most common.

## Management of pathogens in individual rodents and colonies

Treatment of infections in individual rodents, as reported by three quarters of respondents, is expected, being commonplace in veterinary medicine. A third of respondents reported treatment of whole colony infections, which aligns with agricultural practices, such as mass in-water medication in pigs and cattle [44, 45].

Antimicrobials are frequently used to suppress clinical illness or eradicate pathogens from rodent colonies. Published protocols include elimination of *Helicobacter* spp. infection with medicated feed containing amoxicillin, clarithromycin, and metronidazole; elimination of *Corynebacterium bovis* in immune deficient mice with in-water amoxicillin clavulanate; enrofloxacin suppression of *Rodentibacter pneumotropica* infection; and reduction of colony morbidity caused by *Pneumocystis* spp. infection with trimethoprim/sulfamethoxazole [31, 46–49].

## Much antimicrobial use is unjustified

This survey reports frequent prophylactic use around the time of surgery, and in immune-compromised rodent colonies. Infection can be avoided in these cases with institution of strict aseptic practice and husbandry, the latter comprising specialised handling of mice, in laminar flow cabinets with sterile gloves, and provision of sterilised food, bedding, acidified water etc.

Peri-surgical administration of antimicrobials has been considered 'the single most frequently abused principle of veterinary surgery' [25]. Recent retrospective studies in laboratory rodents have found no difference in post-operative infection rates in those animals not administered prophylactic antimicrobials when there is adherence to aseptic technique [32, 50]. Literature in human medicine demonstrates that antimicrobial prophylaxis is unnecessary [51]. With exceptions, such as orthopaedic, neurological, implant and gastrointestinal surgery, surgeons are encouraged to avoid routine perioperative antibiosis [52, 53].

Opportunistic infections in immune compromised mice may be prevented with aseptic husbandry techniques, and use of sterile bedding, food, and acidified or sterilised water [36].

Routine antibiotic administration in immune deficient animals may result in superinfection, as evidenced in nude rats administered months of antimicrobials, of multiple classes, to suppress *Rodentibacter* infection. These rats had never tested positive to *Klebsiella pneumoniae*, yet this pathogen arose within the colony and was extensively drug resistant following antibiosis [54]. The *Rodentibacter* infection recurred once antimicrobials were discontinued.

## The majority of reported antimicrobial use is delivering subtherapeutic doses

The administration of insufficient antibiotic doses, being subtherapeutic for treating infection, induces antibiotic resistance [55]. In rodent facilities, decision on the route of administration is often influenced by labour requirements [56]. Handling of individual rodents when it is necessary to administer medications to a whole colony is time consuming, expensive and requires technical expertise and attention to aseptic husbandry [57]. Consequently, in-water administration is the commonly described route in publications and was reported by most survey respondents.

Such mass treatment is problematic because it cannot be assured that all animals will consume sufficient water to receive an appropriate dose of antimicrobial. Also, many antimicrobials are unstable in water, or when exposed to light, including doxycycline and enrofloxacin [58, 59]. Fewer than half of survey respondents covered water bottles containing these antimicrobials, thus reducing the delivered dose.

This survey also documented a high prevalence of cefovecin administration by injection. Therapeutic plasma concentrations in mice are unachievable with cefovecin, as the drug's half-life in mice is 50 minutes, compared to 7–14 days for cats and dogs, for whom it is registered [60].

## Reported use may represent a risk to human and rodent health

Numerous studies in rodents have found emergence of resistant gut microbiota, even with a single class and dose of antimicrobial [61]. These data suggest that oral administration of antimicrobials, in laboratory rodents, have pronounced and long-standing effects on resistance gene amplification and *de novo* development in gut microbiota [62, 63].

Microbial cross-resistance to multiple antimicrobial classes, conferred by a common molecular mechanism, may arise following administration of a single class [64, 65]. Administration of a broad range of antibiotic classes, as reported in this survey, is likely inducing multi-species microbial cross-resistance.

Resistant microbes are transferred vertically, from rodent dams to offspring during parturition and nursing. Resistance selected for in the microbiota, by antimicrobial administration, may be transmitted indefinitely through generations of animals, as rodent pups acquire their microbiome from their dam and the environment [66, 67]. Mouse-adapted *S. aureus* has been vertically transmitted in mouse colonies for generations [68].

There are numerous examples of cross infection between methicillin resistant *S. aureus* (MRSA) between humans and house mice and rats, and recent documentation of cross-transmission of pathogenic and multidrug resistant (MDR) resistant *E. coli* between humans and companion animals [69–72].

A study across German research animal facilities compared microbiome profiles from the intestines of research rodents, and the skin of personnel working with these animals. Genomic sequencing found that shed skin or dust particles carried by either animals, care takers, or scientists influenced and was shared by the gastrointestinal microbiota of research mice. This demonstrated that despite standard biosecurity engineering controls of humans wearing

masks, gowns and gloves in the facility, there is a significant transfer and sharing of skin microbiota from humans to research rodents. It is plausible that the converse applies, and the rodent microbiome from the gut or skin, can transfer to humans [23, 24, 73]. Another study isolated a human strain of MDR *C. difficile* from an outbreak in a mouse facility, co-located with a human hospital. The authors proposed that the AMR accrued through rodent exposure to antimicrobials and horizontal gene transfer, from commensals to the *Clostridia*. The outbreak was facility wide, despite the work occurring in rooms with high level biosecurity [74, 75].

These examples underscore the fact that laboratory rodents and facilities may harbour dangerous microbes, functioning as a reservoir for resistant pathogens and genes. This is a particular issue in human healthcare settings, and half of survey respondents reported their co-location with human hospitals.

Induction of resistance in rodent pathogens, through unnecessary and subtherapeutic dosing, will also compromise animal welfare, rendering bacterial infections more difficult to treat.

These factors are important to consider given the rodent housing systems documented by respondents. Specifically, conventional open-top cages were common across all facilities, and these generate significant particulate contamination of air, which is then shared between rodent cages, and personnel working the facilities are exposed to particulates, through direct contact and inhalation. During cage washing of PC1 and PC2 cages, staff are likewise, exposed to these aerosols. Sharing of air and particulates is an established work health and safety issue in all rodent facilities below PC3 biocontainment level. Lab animal allergy is a proxy. This is a condition arising from unavoidable exposure of personnel to particulate matter (rodent dander) in the air, via inhalation or direct contact, regardless of cage types and engineering controls [76].

## Disposal of antimicrobials in laboratory rodent waste is causing environmental contamination which may contribute to AMR

This survey showed that antimicrobial containing water is disposed of untreated by most facilities. A third of facilities administering medicated chow dispose of it untreated, in regular municipal waste, where it may contaminate the environment and be consumed by animals. Some facility waste substrates, including bedding of antimicrobial treated animals, containing urine and faeces, also are disposed of without treatment. Such practices risk contaminating the environment with antimicrobials, and microbes that may contain ARGs.

The spread of drug resistance genes has been classified as new type of environmental pollutant [77]. Microbial exposure to anthropogenic antimicrobials in wastewater, agricultural settings, or the built environment, may select for AMR in the environmental resistome, generating reservoirs of resistant pathogens and microbial reservoirs of ARGs [78–80].

Hospital and laboratory wastewater are established contributors to AMR and release of ARGs to the environment, underscoring the need for efforts to treat water pre-disposal [81, 82]. Recent reviews have identified the significant and overlooked contribution of inappropriately disposed of antimicrobials as drivers of AMR in both human and veterinary medicine [83]. Whilst there is some guidance around disposal of laboratory waste that contains antimicrobials, there are no equivalent publicly available recommendations for laboratory *animal* waste containing antimicrobials [84].

Antimicrobials commonly reported in the survey, have long-standing environmental impacts. Whilst antimicrobial concentrations in the environment consequent to use in rodents are unknown, as they have not been measured, those from hospital wastewater discharge have. Numerous studies quantifying these demonstrate that discharge of stable antimicrobials is

inducing resistance in environmental microbes. These microbes inherently possess a significant diversity of ARG, however, the release of sublethal antimicrobials exerts a substantial selective pressure resulting in acquisition of ARGS. These resistant pathogens may be transmitted back to animals and humans via direct contact with the environment, food, and drinking water [85].

Enrofloxacin is the most administered antimicrobial, reported by this survey's population, and is a stable antimicrobial pollutant with a half-life as long as 3–9 years in natural environments [86]. Hospital effluent, into which much rodent waster water is reportedly discharged, has been established as a significant source in the spread of fluoroquinolones, such as enrofloxacin, into the environment [87].

Likewise, survey respondents reported a high prevalence of tetracycline use. A high percentage (50–90%) of administered tetracyclines are excreted via an animal's urine and faeces unchanged, meaning they are discharged into the environment in rodent excreta through bedding disposal, as well as in chow, and wastewater discharge. Tetracyclines have an environmental half-life of up to 120 days [88]. Birds and rodents consuming tetracycline containing chow and bedding from landfill may disseminate antimicrobial resistant microbes and their ARGs to human and other animal populations [89].

Recent studies document the aerosolization and dissemination of resistant microbes and ARGs during routine passage of waste through transfer stations, risking the health and safety of waste workers and people living proximal to municipal landfill [90, 91]. There is also concern regarding the spread of antimicrobials, and associated resistant microbes and ARGs will transfer beyond landfill via leachate and landfill leakage [87].

## Conclusion

In summary, this survey characterises and quantifies, for the first time, the extent of the use of antimicrobials in research rodent facilities in Australia and New Zealand. The data align with laboratory rodent literature describing regimens and protocols for antimicrobial use within research projects. We believe this is the first published quantitative, and qualitative survey of its type, globally, characterising prevalence of use, indications for and regimens of administration, means of access, and common modes of disposal of antimicrobials, administered to laboratory rodents.

The survey identified areas where antimicrobial use is not indicated, and is injudicious, as well as confirming its widespread use, inherent to rodent research models, as researchers apply published consensus antimicrobial protocols. The imprudent use reported, including commonplace subtherapeutic dosing and administration of antimicrobials of critical importance to human health, and inappropriate disposal, is likely contributing to the emergence of AMR in laboratory rodents, in the environment and potentially, in persons working with the animals. This is particularly concerning where the rodent facility is co-located with a hospital since there will be many opportunities for transfer on personnel who move between the animal facility and the hospital if inadvertent breaches of biosecurity occur.

Inappropriate or imprudent use of antimicrobials also has negative implications for the research rodents, which may develop untreatable infections, as well as impacts for public health at a facility and environmental level. The survey has identified an urgent need to develop and implement evidence-based standard operating principles for responsible antimicrobial usage and disposal in rodent research facilities, in Australia and New Zealand.

## Supporting information

**S1 Dataset. Survey questions and responses.**
(XLSX)

**S1 Text. Inclusivity in global research questionnaire.**
(DOCX)

## Acknowledgments

We would like to express our gratitude to the laboratory animal community in Australia and New Zealand, for their voluntary participation in the survey. We also wish to thank the ANZLAA Secretariat for providing the contact details of all laboratory animal facilities in these countries. We would also like to acknowledge the support provided by the Statistical Consulting Centre, The University of Melbourne, for assistance with data analysis.

## Author Contributions

**Conceptualization:** Rebbecca S. Wilcox.

**Data curation:** Rebbecca S. Wilcox.

**Formal analysis:** Rebbecca S. Wilcox, Marc S. Marenda, Joanne M. Devlin, Colin R. Wilks.

**Investigation:** Rebbecca S. Wilcox.

**Methodology:** Rebbecca S. Wilcox.

**Supervision:** Marc S. Marenda, Joanne M. Devlin, Colin R. Wilks.

**Visualization:** Rebbecca S. Wilcox.

**Writing – original draft:** Rebbecca S. Wilcox.

**Writing – review & editing:** Marc S. Marenda, Joanne M. Devlin, Colin R. Wilks.

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
