## [Decision Letter · Decision Letter 0]

5 Mar 2024

PONE-D-23-30986Antimicrobial use in laboratory rodent facilities in Australia and New Zealand- a cross-sectional survey of veterinarians and facility managersPLOS ONE

Dear Dr. Wilcox,

Thank you for submitting your manuscript to PLOS ONE. After careful consideration, we feel that it has merit but does not fully meet PLOS ONE’s publication criteria as it currently stands. Therefore, we invite you to submit a revised version of the manuscript that addresses the points raised during the review process.

Please add the point made by the reviewer in the discussion 

We look forward to receiving your revised manuscript.

Kind regards,

Iddya Karunasagar

Academic Editor

PLOS ONE

Journal Requirements:

3. Please include a complete copy of PLOS’ questionnaire on inclusivity in global research in your revised manuscript. Our policy for research in this area aims to improve transparency in the reporting of research performed outside of researchers’ own country or community. The policy applies to researchers who have travelled to a different country to conduct research, research with Indigenous populations or their lands, and research on cultural artefacts. The questionnaire can also be requested at the journal’s discretion for any other submissions, even if these conditions are not met.  Please find more information on the policy and a link to download a blank copy of the questionnaire here: https://journals.plos.org/plosone/s/best-practices-in-research-reporting. Please upload a completed version of your questionnaire as Supporting Information when you resubmit your manuscript.

4 We note that your Data Availability Statement is currently as follows: "All relevant data are within the manuscript"

If there are ethical or legal restrictions on sharing a de-identified data set, please explain them in detail (e.g., data contain potentially sensitive information, data are owned by a third-party organization, etc.) and who has imposed them (e.g., an ethics committee). Please also provide contact information for a data access committee, ethics committee, or other institutional body to which data requests may be sent. If data are owned by a third party, please indicate how others may request data access

Additional Editor Comments:

Please see comments from Reviewer on discussion section

Reviewers' comments:

Reviewer's Responses to Questions

**Comments to the Author**

1. Is the manuscript technically sound, and do the data support the conclusions?

Reviewer #1: Yes

2. Has the statistical analysis been performed appropriately and rigorously? 

Reviewer #1: Yes

3. Have the authors made all data underlying the findings in their manuscript fully available?

Reviewer #1: Yes

4. Is the manuscript presented in an intelligible fashion and written in standard English?

Reviewer #1: Yes

5. Review Comments to the Author

Reviewer #1: Most of the animal facilities dispose the biomedical waste as per the local rules. Hence, residual effect of the antibiotics in environment needs to be elaborated considering the half-life of the drug, concentration etc., in the Discussion section.

6. PLOS authors have the option to publish the peer review history of their article (what does this mean?). If published, this will include your full peer review and any attached files.

Reviewer #1: No

---

## [Author Response · Author response to Decision Letter 0]

17 Apr 2024

18/04/2024

To whom it may concern,

We are delighted with significance of Reviewer #1’s comment: “Most of the animal facilities dispose the biomedical waste as per the local rules. Hence, residual effect of the antibiotics in environment needs to be elaborated considering the half-life of the drug, concentration etc., in the Discussion section.” Environmental contamination is one of the most concerning findings of this study.

Within the Discussion section, we have elaborated on this matter, including the specific environmental half-lives of the most used antimicrobials in laboratory rodents, according to this survey’s respondents. The concentrations in the environment are unknown, from this source, as there are no data or studies that can be found, and we have acknowledged this knowledge gap. We have also expanded on the issue as a newly acknowledged driver of antimicrobial resistance, and environmental pollution, along with the pathways by which human and animal health may be affected by these disposal practices. 

We have completed and attached the Inclusivity in Global Research Questionnaire and amended both the tracked and clean manuscripts to reflect requested changes, and uploaded these.

All changes have been referenced with peer reviewed literature.

Thank you for your consideration,

Dr Wilcox

---

## [Editor Report · Decision Letter 1]

25 Apr 2024

Antimicrobial use in laboratory rodent facilities in Australia and New Zealand- a cross-sectional survey of veterinarians and facility managers

PONE-D-23-30986R1

Dear Dr. Wilcox,

We’re pleased to inform you that your manuscript has been judged scientifically suitable for publication and will be formally accepted for publication once it meets all outstanding technical requirements.

Kind regards,

Iddya Karunasagar

Academic Editor

PLOS ONE

Additional Editor Comments (optional):

All comments have been addressed.
---

## [Editor Report · Acceptance letter]

13 Jun 2024

PONE-D-23-30986R1 

PLOS ONE

Dear Dr. Wilcox, 

I'm pleased to inform you that your manuscript has been deemed suitable for publication in PLOS ONE. Congratulations! Your manuscript is now being handed over to our production team.

Kind regards, 

on behalf of

Dr. Iddya Karunasagar 

Academic Editor

PLOS ONE